# Radiomics in Dermatological Optical Coherence Tomography (OCT): Feature Repeatability, Reproducibility, and Integration into Diagnostic Models in a Prospective Study

**DOI:** 10.3390/cancers17050768

**Published:** 2025-02-24

**Authors:** Yousif Widaatalla, Tom Wolswijk, Muhammad Danial Khan, Iva Halilaj, Klara Mosterd, Henry C. Woodruff, Philippe Lambin

**Affiliations:** 1The D-Lab, Department of Precision Medicine, Maastricht University, 6200 MD Maastricht, The Netherlands; danial.khan.guest@maastrichtuniversity.nl (M.D.K.); i.halilaj@maastrichtuniversity.nl (I.H.); h.woodruff@maastrichtuniversity.nl (H.C.W.); philippe.lambin@maastrichtuniversity.nl (P.L.); 2GROW Research Institute for Oncology and Reproduction, Maastricht University, 6200 MD Maastricht, The Netherlands; tom.wolswijk@mumc.nl (T.W.); k.mosterd@mumc.nl (K.M.); 3Department of Dermatology, Maastricht University Medical Center+, 6229 HX Maastricht, The Netherlands; 4Department of Radiology and Nuclear Medicine, Maastricht University Medical Center+, 6229 HX Maastricht, The Netherlands

**Keywords:** artificial intelligence, radiomics, test–retest, OCT

## Abstract

This study investigates the potential of radiomics in dermatological optical coherence tomography (OCT) by evaluating the repeatability and reproducibility of handcrafted radiomics features (HRFs) under a test–retest protocol for varying intensity quantization bin widths. From 80 OCT scans of 20 volunteers, six stable HRFs were identified across all bin widths, with a width of 25 emerging as the optimal choice for balancing stability and textural detail. A multiclass classifier trained on these robust features demonstrated stable performance in differentiating between different lesions compared to recursive feature elimination, highlighting the value of incorporating robust and stable features. These findings establish a framework for integrating radiomics into dermatological diagnostics, advancing toward a non-invasive alternative to traditional biopsies.

## 1. Introduction

With more than 300,000 cases of skin cancer diagnosed in 2022, early detection and diagnosis remain a challenge in dermatology [1]. Optical coherence tomography (OCT) has recently gained popularity in skin imaging as a non-invasive replacement for biopsy, showing high diagnostic accuracy for diagnosing equivocal lesions suspected of basal cell carcinoma (BCC) [2,3]. Using near-infrared light interferometry, OCT enables the visualization of the gross architecture of the skin and adnexal structures with a depth of approximately 1.5 mm [4,5]. Like with many new technologies, training healthcare specialists to read OCT scans is a time-consuming task, and diagnosis suffers from subjectivity, leading to inter- and intra-observer variability [6].

Radiomics, the application of machine learning and deep learning to medical images, could help increase diagnostic robustness and speed. Radiomics relies on the extraction of repeatable and reproducible quantitative data from medical images, offering significant potential to guide clinical decision-making [7,8,9]. Applications such as tumor phenotyping are widely established in research using imaging modalities such as computed tomography (CT) and magnetic resonant imaging (MRI) [10,11,12]. However, the potential of radiomics in OCT remains less explored compared to these traditional modalities.

To develop a radiomics-based model, handcrafted radiomics features (HRFs) are extracted from a manually or automatically segmented region of interest (ROI) by applying mathematical formulas to the array of intensity values [13,14]. These features can be mined directly from the original image or from images modified by applying filters such as the Laplace transform of Gaussian or wavelet decomposition. HRFs are categorized into groups depending on their characteristics, e.g., ‘first-order features’ which detail the distribution of intensities, ‘shape features’ detailing the geometry of the ROI, or ‘texture features’ describing the relations between pixel intensities [15]. These features can be used to train predictive models that support diagnosis or follow-up without the need for invasive medical intervention [16].

Although radiomics, including handcrafted features and deep learning, has seen significant advancements with over 700 AI algorithms approved by the Food and Drug Administration (FDA) [17], challenges such as reproducibility, standardization, and integration into clinical workflows continue to limit its widespread adoption in routine clinical practice [18]. In dermatology, for instance, the lack of standardized systems for storing and sharing dermatological images, unlike the Picture Archiving and Communication System (PACS) used in radiology, limits data interoperability [19].

Another major issue is feature stability, as some radiomics features are highly sensitive to slight variations in image acquisition, reconstruction parameters, or preprocessing steps [20]. Therefore, test–retest studies are essential for identifying robust HRFs. These robust HRFs can then be used to develop a radiomics model that is reliable and applicable in various clinical settings [21].

Recent studies have highlighted the potential of radiomics-based models in various skin conditions. For instance, Turani et al. developed a radiomics classifier to differentiate melanoma from benign nevi, achieving high sensitivity and specificity rates of 97% and 98%, respectively [22]. Similarly, Marvdashti et al. investigated the automated detection of BCC using polarization-sensitive OCT, reporting an area under the receiver operating characteristic curve (AUC) of 0.97 and an accuracy of 95% [23].

Dermatological OCT is a promising emerging modality for such models and as it is increasingly integrated into routine clinical practice [24], the demand for reliable and robust models continues to grow. Also in OCT, a critical step in achieving generalizability is the identification of robust HRFs [25].

To address this gap, our study is the first to systematically evaluate the repeatability and reproducibility of HRFs from OCT images of benign nevi under test–retest conditions. We also assess the impact of intensity binning strategies on HRF stability and demonstrate their clinical utility by incorporating these features into a classifier. Ultimately, this work establishes a framework for enhancing predictive model performance in dermatology.

## 2. Materials and Methods

### 2.1. Study Design and Volunteers

This test–retest prospective study was conducted at the Department of Dermatology of Maastricht University Medical Center+ (MUMC+). Healthy volunteers aged 18 years or older with at least two benign nevi were included, while participants who declined to sign informed consent were excluded. Our main objective was to assess the robustness of HRFs in OCT scans. Therefore, we focused on evaluating feature stability under controlled conditions using easily accessible lesions, such as benign nevi. This study was reviewed by the local medical ethical committee and registered on clinicaltrial.gov (NCT05517889). The study workflow is presented in Figure 1.

### 2.2. OCT Imaging and Data Acquisition

Each participant underwent two consecutive scans for each nevus with a standardized 10 min interval between scans to ensure a consistent timeframe for image acquisition. This balance accommodated the temporal dynamics of the imaging modality while ensuring consistency and comparability across our study. A medical doctor (TW) with OCT experience performed the scanning using the Vivosight Multi-beam Swept-Source Frequency Domain OCT (Michelson Diagnostics, Maidstone, Kent, UK). The OCT device specifications included a class 1 eye-safe laser, lateral resolution of <7.5 µm, axial resolution of <5 µm, depth of focus of up to 1.5 mm, and scan area of 6 × 6 mm^2^. All scans were co-registered and exported from the device in Digital Imaging and Communications in Medicine (DICOM) format for analysis.

For the classification task, we utilized an additional MUMC+ dataset, which included histopathologically confirmed cases of BCC (*n* = 63) and Bowen’s disease (*n* = 31). This dataset was obtained from a prospective cohort study at the dermatology outpatient clinic of MUMC+, including patients who underwent skin biopsy for suspected non-melanoma skin cancer or premalignancy. Further details can be found in the work of Sinx et al. [26]. Furthermore, we incorporated scans of benign nevi (*n* = 40) from our current repeatability and reproducibility study. In total, 134 OCT scans were used to train and validate the radiomics classifier.

### 2.3. Image Preprocessing and Segmentation

For both datasets, all DICOM files were uploaded to ITK_SNAP software (version 3.8.0, Philadelphia, PA, USA) [27] and a medical doctor (YW) manually segmented all the nevi. Subsequently, another medical doctor (TW), with extensive experience in OCT assessment, confirmed all segmentations and corrections were made by consensus if necessary. All images were saved in nearly raw raster data (NRRD) format, which is widely used in medical image analysis, as it simplifies processing and anonymizes patient data by removing DICOM metadata.

Gray-level quantization, an essential step in radiomics analysis, involves discretizing intensity values into distinct bins before extracting HRFs [28] to reduce noise and computation time. Two common methods are a fixed bin size (FBS) and fixed bin number (FBN) [29]. This study uses a FBS, referred to as bin width (BW). While the default PyRadiomics (version 2.2.0) BW of 25 works well for broader intensity ranges like those in CT (−1000 to +1000 Hounsfield units) [30,31], the narrower intensity range of OCT (0–255) may require further investigation to determine an optimal BW [32]. Selecting an appropriate BW simplifies data by grouping similar intensities while preserving key features. To assess the impact of BW on feature stability, we compared BWs of 5, 10, 15, 25, 30, 35, 40, 45, and 50. Given that all images had the same spatial resolution, we did not investigate the role of resampling in this study.

### 2.4. HRFs Extraction

Features were extracted using the PyRadiomics open-access feature extraction package (version 2.2.0) [33]. Before extracting HRFs, although monochromatic was saved in a file format containing three channels (red, green, and blue), each scan was transformed back into a grayscale array. HRFs were extracted from the 3D ROI using different BWs in the preprocessing step to allow for the assessment of the HRFs’ repeatability and reproducibility.

The extracted HRFs were divided into the following groups: first-order statistics; shape features; and texture features, which could be divided further into gray-level run-length matrix (GLRLM), neighborhood gray-tone difference matrix (NGTDM), gray-level size-zone matrix (GLSZM), gray-level co-occurrence matrix (GLCM), and gray-level dependence matrix (GLDM) features.

### 2.5. Statistical Analysis

#### Repeatability and Reproducibility of HRFs

At each BW, the concordance correlation coefficient (CCC) was calculated between the test and retest scans to evaluate feature repeatability. For reproducibility, the CCC was calculated between the same scans using different BWs. Features were considered repeatable or reproducible if CCC ≥ 0.9. The CCC was calculated using Python (version 3.9) and is widely used in radiomics test–retest studies due to its ability to assess agreement, detect systematic differences, and account for data variability. This makes it a reliable metric for evaluating radiomics features, which are critical for constructing accurate models [34].$$CCC=\frac{2\rho \sigma_{1}\sigma_{2}}{\sigma_{1}^{2}+\sigma_{2}^{2}+{\left(\mu_{1}-\mu_{2}\right)}^{2}}$$

μ, ρ, and σ represent the mean, correlation coefficient, and variance for the two datasets (test and retest), respectively. CCC values range from −1 (inverse concordance) to 0 (no concordance) and up to 1 (perfect agreement).

Additionally, Spearman’s correlation was performed on the list of repeatable HRFs at each BW to remove highly correlated features. This step minimized redundancy and identified a final set of unique and repeatable HRFs at each BW.

### 2.6. Multiclass Classification

After identifying stable HRFs across different BWs, we developed two radiomics classifiers to distinguish between benign nevi, BCC, and Bowen’s disease. We used the Extreme Gradient Boosting (XGBoost) algorithm (version 2.1.2), a robust machine learning classifier, due to its ability to handle high-dimensional data like HRFs and to prevent overfitting through regularization and early stopping [35]. The classification dataset was split into training (70%) and test (30%) sets.

Given that our dataset was imbalanced, with fewer cases of Bowen’s disease compared to BCC and benign nevi, we performed stratified 5-fold cross-validation to maintain class distribution across folds. Additionally, model performance was evaluated using the AUC and precision and recall metrics, which offer more informative assessments for imbalanced datasets compared to accuracy alone.

We compared the impact of training the model using only the robust features identified in our study to the standard radiomics pipeline, which usually involves the use of feature selection methods such as recursive feature elimination (RFE). We reported the training and test performance of each lesion using a ‘One-vs.-Rest’ (OvR) strategy (e.g., nevi vs. BCC and Bowen’s disease) with precision, recall, F1-score, accuracy, and the AUC, along with their corresponding confidence intervals (CIs). The AUC and other reported metrics were calculated using the classification performance of the radiomics classifiers in Python, utilizing the scikit-learn library (version 1.5.2).

Additionally, a negative control experiment was conducted by randomizing labels once before training to confirm that the model’s performance was based on meaningful patterns rather than chance or overfitting. The test set remained separate throughout, and randomized labels were not altered during iterations.

## 3. Results

### 3.1. Patient Characteristics

For the repeatability and reproducibility parts of this study, a total of 20 healthy volunteers were included, with a mean age of 28 years. A total of 40 benign lesions (nevi) were scanned twice with a ten-minute interval, resulting in 80 scans. Patient characteristics and lesion sites are detailed in Table 1. For the classification task, a total of 134 scans were utilized, comprising BCC (*n* = 63), Bowen’s disease (*n* = 31), and benign nevi (*n* = 40).

### 3.2. Repeatability HRFs Across Bin Widths

We evaluated the repeatability of 107 features, categorized into first-order statistics, shape, and texture features (GLCM, GLRLM, GLSZM, GLDM, NGTDM), across BWs ranging from five to fifty, in increments of five, with results detailed in Table 2.

First-order features showed consistent stability, with seven features achieving CCC ≥ 0.9 (Figure 2B). Shape features had the highest repeatability (85%), with 12 features remaining consistent across all BWs (Figure 2A). It is important to note that shape features remained unaffected by gray-level binning; thus, any observed differences solely reflected variations in lesion shape between test and retest scans. In contrast, texture features exhibited greater sensitivity to BW variations. The GLCM repeatability differed above and below a BW of 25 (Figure 2C). The GLRLM and GLSZM were more repeatable at finer BWs (Figure 2D,E), whereas the GLDM and NGTDM achieved better repeatability at larger BWs, particularly those of 30 and 35 (Figure 2F,G).

#### Non-Correlated Repeatable Feature Across BWs

After removing highly correlated features, six features—four shape features (Elongation, Least Axis Length, Major Axis Length, Surface–Volume Ratio), the ‘First-Order Minimum’, and the ‘GLCM Correlation’—remained consistently stable across all BWs. However, the texture features exhibited variability in stability depending on the BW. The ‘GLCM Maximum Probability’ and ‘GLRLM Long-Run Emphasis’ were stable only at the smallest BW (5), while the ‘GLSZM Low-Gray-Level Emphasis’ and ‘GLDM Small-Dependence Low-Gray-Level Emphasis’ became repeatable at larger BWs (20+). Moreover, features such as the GLRLM ‘Run Entropy’ and ‘GLSZM High-Gray-Level-Zone Emphasis’ were stable exclusively at a BW of 50, highlighting the influence of larger BWs on capturing texture details.

### 3.3. Agreement of HRFs’ Reproducibility Across BW Pairs

Reproducibility assessed how many features remained consistent when the BW changed for the same scan. The highest reproducibility was observed between larger BWs, with 58 features reproducible between BWs of 30 and 40. In contrast, the smallest BWs (5 and 10) had the lowest reproducibility, with only 38 features. Intermediate BWs, such as those of 20 and 25, showed a moderate reproducibility of 53 features (Figure 3).

### 3.4. Multiclass Classifier

We trained our model on 70% of the data with stratified five-fold cross-validation. The classifier (nevi–BCC–Bowen’s disease) trained with six robust features identified in our study achieved a mean AUC of 0.96 (95% CI: 0.93–0.98), while the RFE-based classifier achieved a slightly higher mean AUC of 0.97 (95% CI: 0.95–0.99). Precision, recall, and F1-scores were comparable between the two classifiers, with the RFE-based model showing a slight edge for BCC and Bowen’s disease (Appendix A). We further tested both classifiers on an unseen test set (30% of the dataset). The robust feature-based classifier had a better performance overall, achieving an accuracy of 90%. The AUCs for the robust classifier were 0.96 (95% CI: 0.89–1.00) for BCC, 0.94 (95% CI: 0.87–1.00) for Bowen’s disease, and 1.00 (95% CI: 1.00–1.00) for nevi (Figure 4B, Appendix B—Table A1).

In comparison, the RFE-based classifier demonstrated a lower performance on the test set, with an accuracy of 76%. The per-class AUCs for the RFE model were 0.86 (95% CI: 0.75–0.96) for BCC, 0.80 (95% CI: 0.62–0.89) for Bowen’s disease, and 1.00 (95% CI: 1.00–1.00) for nevi (Figure 4A, Appendix B—Table A2)

To ensure that our results were not due to chance or overfitting, a negative control experiment was performed by randomizing the labels and re-training the model. The ROC-AUC curves are in Appendix C.

## 4. Discussion

To our knowledge, this study is the first to prospectively investigate the repeatability and reproducibility of HRFs extracted from OCT scans, alongside the impact of BW selection on feature stability. This is particularly relevant as recent publications have identified the lack of external generalizability as a primary barrier to translating radiomics models into clinical practice [36,37,38,39]. To address this, we examined the stability of HRFs in OCT in a prospectively collected test–retest dataset and constructed radiomics models to demonstrate how incorporating stable HRFs influenced model performance. Our results illustrate that the choice of BW affected the number of repeatable and reproducible HRFs, thereby impacting the performance of the radiomics model.

OCT, in comparison to CT and MRI, presents unique challenges in radiomics analysis. While CT benefits from standardized HU and MRI is affected by scanner-specific intensity variations, OCT lacks an established intensity normalization standard. Moreover, whereas CT and MRI radiomics features are influenced by slice thickness and reconstruction settings, OCT features are more sensitive to probe pressure, patient motion, and signal attenuation. These differences highlight the need for further research to standardize OCT radiomics and improve feature reproducibility for clinical applications.

Since the shape features were not susceptible to BW, they were the most repeatable group, with 85% of the features achieving a CCC value ≥ 0.9. However, two shape features, ‘Original Shape Flatness’ and ’Original Shape Sphericity’, were not repeatable (CCC < 0.9). This can be attributed to the sensitivity of the shape features to the geometry of the segmented lesions, which may have varied due to slight differences in patient positioning and probe pressure during consecutive OCT scans. These findings align with prior studies in CT and MR imaging modalities [40,41,42,43], which similarly observed that shape features exhibit high repeatability but are susceptible to acquisition-related variability.

The texture features demonstrated considerable variability in repeatability depending on the BW. For example, the GLCM features were more repeatable with smaller bin widths, whereas the GLRLM features showed improved repeatability with larger bin widths. This variability reflects the nature of the texture features, which captured diverse aspects of spatial and intensity distributions within the scan. Similar trends were reported by Doumou et al., who studied the effect of BW choice on texture features in PET imaging [44], and Shafiq-ul-Hassan et al., who found that only a limited subset of texture features remained stable across gray-level discretization schemes in CT imaging [45].

To refine the analysis, Spearman’s correlation was applied to eliminate redundant features, yielding a list of unique HRFs at each BW. Among these, six features were consistently present across all BWs. Beyond these consistent features, the stability of other features was influenced by the choice of bin width. For example, the ‘GLCM Maximum Probability’ and ‘GLRLM Long-Run Emphasis’ were stable only at the smallest bin width (BW of 5), whereas the ‘GLSZM Low-Gray-Level Emphasis’ and ‘GLDM Small-Dependence Low-Gray-Level Emphasis’ were repeatable at larger bin widths (BW ≥ 20). Features like the ‘GLRLM Run Entropy’ and ‘GLSZM High-Gray-Level-Zone Emphasis’ became repeatable exclusively at the largest bin width (BW 50), indicating that broader gray-level discretization captured additional textural details. Based on these findings, and as shown in Table 2, a BW of 25 is recommended for retaining repeatable features while capturing meaningful textural details.

Several studies have explored the repeatability of HRFs through test–retest analyses in phantom and vivo settings in other imaging modalities [46,47]. For instance, Granzier et al. examined the repeatability of HRFs extracted from MRI breast scans using three different imaging settings: the T1-weighted (T1W), T2-weighted (T2W), and apparent diffusion coefficient (ADC) settings [48]. Their findings demonstrated that the repeatability of radiomics features varied across these settings. Similarly, Li et al. investigated the reproducibility of MRI-derived HRFs in the hippocampus in conditions like Alzheimer’s disease, cognitive impairment, and multiple sclerosis [49]. The authors observed that texture features from the hippocampus exhibited high stability, making them a valuable neuroimaging biomarker.

These findings emphasize the importance of selecting HRFs that are repeatable and non-redundant for robust radiomics modeling. The variability observed across BWs highlights the need for careful preprocessing and feature selection to mitigate the effects of BW sensitivity, ultimately improving the generalizability of radiomics models.

Our results demonstrate that the number of reproducible HRFs varied across BW pairs. The higher reproducibility observed with larger BWs can be attributed to their ability to smooth out variations in intensity values, effectively reducing the impact of noise and artifacts in the scans. This smoothing results in a more stable extraction of textural features, at the price of removing possible nuances in information contained within the image. Conversely, smaller BWs introduce finer granularity, making features more sensitive to minor intensity variations, thereby reducing their reproducibility.

Our findings align with the work of Ying Li et al., who observed the highest proportion of reproducible features with larger gray-level bin widths, such as a BW of 50 in CT scans. However, they cautioned against using such large BWs due to volume-confounding effects [50]. Similarly, Larue et al. reported that in CT scans, while feature values consistently changed with BW variation, very small or very large BWs resulted in wide variability in feature values [51]. Furthermore, they found that a BW of 25 produced more consistent and comparable feature values.

Our classification results highlight the important role of incorporating robust HRFs in improving model performance and generalizability. When comparing the performance of the conventional radiomics approach, which involved selecting HRFs through RFE, to our method of using robust HRFs, a clear improvement was seen in accuracy and the AUCs. The impact of using robust features has also been observed in previous radiomics studies. For instance, Teng et al. developed and compared radiomics models for head and neck cancer using all features versus robust features only and noted a significant improvement in the models’ generalizability and robustness [52]. Moreover, in the negative control experiment, the results demonstrated no meaningful predictive performance, further supporting the validity of our findings. These findings emphasize the critical role of selecting stable features in building robust radiomics classifiers.

Our study has a few limitations. First, the small sample size of 20 volunteers, 40 nevi, and 80 scans may limit the generalizability of our findings. As our study establishes a framework for assessing feature robustness, external validation with larger datasets is needed to confirm reproducibility across different imaging settings and patient populations. Furthermore, we believe that prospective validation in real clinical cohorts is essential to evaluating the practical impact of incorporating OCT radiomics into routine dermatological workflows. Second, this study was conducted at a single center, which restricted our ability to perform reproducibility analysis with scans from different devices, thereby limiting the broader applicability of our results. Third, the classifier was developed using a dataset of 134 patients representing only three conditions, which may not have fully reflected the clinical variability encountered in larger and more diverse populations. Fourth, while our manual segmentation was validated by two assessors, it remained operator-dependent and may have introduced variability. Semi-automatic methods could improve reproducibility; however, validated tools for OCT are limited. Future studies should explore automated approaches to reduce inter-observer variability. Fifth, Laser Speckle in OCT scans can cause local intensity variations, which may affect the stability of HRFs across different bin widths. This potential impact should be addressed in future studies through noise reduction techniques or further investigation into its effect on feature stability. Finally, the exclusion of filter-based features due to their complexity and low repeatability means that our findings might not have captured all the features that could have influenced the model, highlighting the need for future research to improve the handling of these complex features.

## 5. Conclusions

Our study highlights the potential of HRFs as non-invasive diagnostic markers and emphasizes the critical role of BW selection in enhancing HRF stability. Based on our findings, we recommend using a bin width of 25, as it balances the retention of stable features across various groups while capturing meaningful textural details. By establishing a methodological framework for optimizing preprocessing steps, this work lays the foundation for improving the stability and predictive accuracy of OCT radiomics models. The identification of robust HRFs not only supports the development of accurate predictive, diagnostic, and prognostic models but also advocates their integration into future decision-support systems. Ultimately, this approach could be an improved, promising, non-invasive alternative to traditional biopsies in dermatology, advancing the clinical relevance of radiomics research.

## Figures and Tables

**Figure 1 cancers-17-00768-f001:**
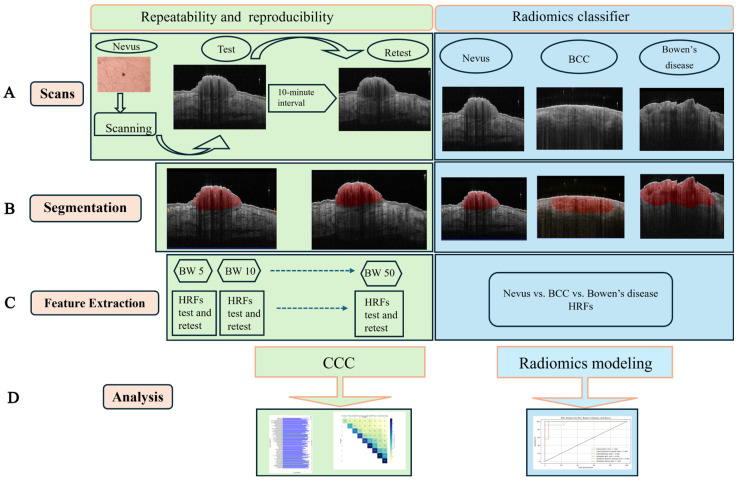
The workflow of this study and key steps: (**A**) Dataset acquisition. (**B**) Scan delineation. (**C**) Feature extraction, including intensity, shape, and texture features. (**D**) Repeatability and reproducibility assessment using the concordance correlation coefficient (CCC ≥ 0.9). Stable hand-crafted radiomics features (HRFs) were subsequently used to develop a multi-class radiomics classifier for distinguishing basal cell carcinoma (BCC), Bowen’s disease, and nevi. The workflow also evaluates the impact of bin width (BW) on feature stability and classifier performance.

**Figure 2 cancers-17-00768-f002:**
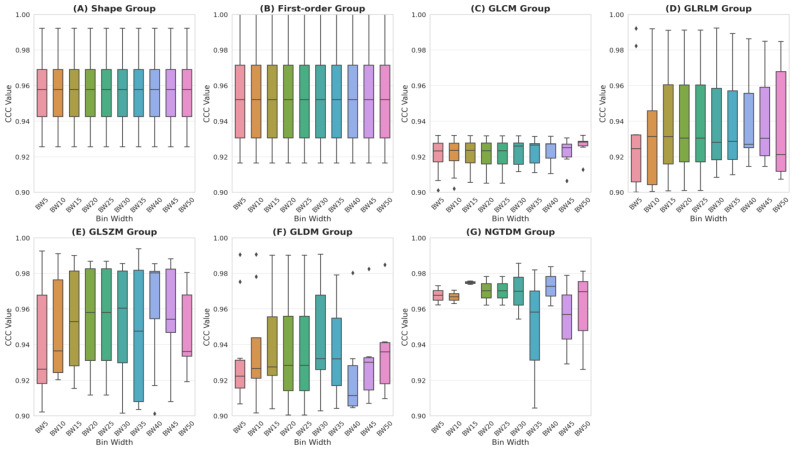
Box plots of CCC values across bin widths (BWs) for different HRF groups. Shape features remain unaffected by gray-level binning, with variations reflecting differences in lesion shape between test and retest scans. Each box plot represents median (line), interquartile range (box), and full range of values (whiskers).

**Figure 3 cancers-17-00768-f003:**
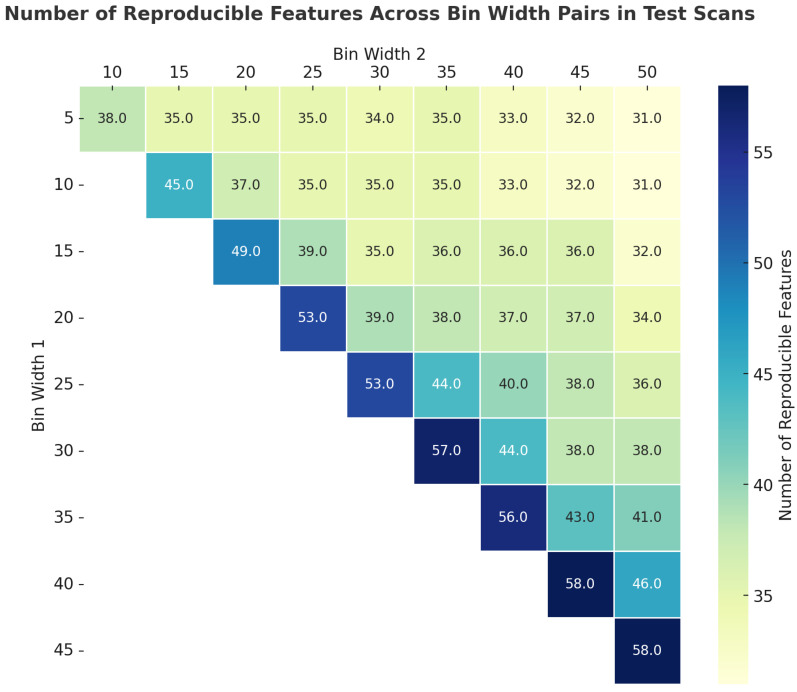
Pairwise (per scan) comparison of reproducibility of HRFs across different BWs.

**Figure 4 cancers-17-00768-f004:**
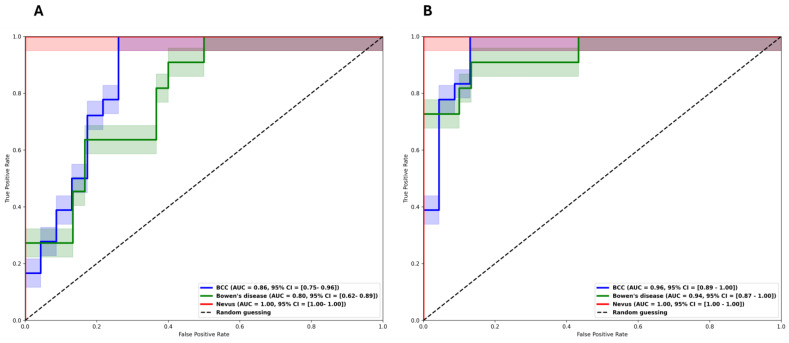
Radiomics classification results (**A**) without pre-selected repeatable HRFs and (**B**) with robust features only.

**Table 1 cancers-17-00768-t001:** Participant characteristics and lesion sites.

Characteristic	(Mean ± SD)/n (/%)
Age (years)	28 ± 7
Sex (*n* = 20)	
Men	4 (20.0%)
Women	16 (80.0%)
Fitzpatrick skin type (*n* = 20)	
Type 1	7 (35.0%)
Type 2	10 (50.0%)
Type 3	2 (10.0%)
Type 4	1 (5.0%)
Type 5	0 (0%)
Type 6	0 (0%)
Localization (*n* = 40)	
Head or neck	7 (17.5%)
Extremities	24 (60.0%)
Trunk	9 (22.5%)

**Table 2 cancers-17-00768-t002:** List of unique repeatable HRFs per BW after applying Spearman’s correlation to remove redundant features, showing retained non-correlated features across different BWs.

Bin Width	Feature Name
5	Shape Elongation
	Shape Least Axis Length
	Shape Major Axis Length
	Shape Maximum 2D Diameter Row
	Shape Surface–Volume Ratio
	First-Order Minimum
	GLCM Correlation
	GLCM Maximum Probability
	GLRLM Long-Run Emphasis
10	Shape Elongation
	Shape Least Axis Length
	Shape Major Axis Length
	Shape Maximum 2D Diameter Row
	Shape Surface–Volume Ratio
	First-Order Minimum
	GLCM Correlation
15	Shape Elongation
	Shape Least Axis Length
	Shape Major Axis Length
	Shape Maximum 2D Diameter Row
	Shape Surface–Volume Ratio
	First-Order Minimum
	GLCM Correlation
20	Shape Elongation
	Shape Least Axis Length
	Shape Major Axis Length
	Shape Maximum 2D Diameter Row
	Shape Surface–Volume Ratio
	First order Minimum
	GLCM Correlation
	GLSZM Low-Gray-Level-Zone Emphasis
25	Shape Elongation
	Shape Least Axis Length
	Shape Major Axis Length
	Shape Maximum 2D Diameter Row
	Shape Surface–Volume Ratio
	First-Order Minimum
	GLCM Correlation
	GLSZM Low-Gray-Level-Zone Emphasis
	GLDM Small-Dependence Low-Gray-Level Emphasis
30	Shape Elongation
	Shape Least Axis Length
	Shape Major Axis Length
	Shape Maximum 2D Diameter Row
	Shape Surface–Volume Ratio
	First-Order Minimum
	GLCM Correlation
	GLSZM Low-Gray-Level-Zone Emphasis
35	Shape Elongation
	Shape Least Axis Length
	Shape Major Axis Length
	Shape Maximum 2D Diameter Row
	Shape Surface–Volume Ratio
	First-Order Minimum
	GLCM Correlation
	GLSZM Low-Gray-Level-Zone Emphasis
40	Shape Elongation
	Shape Least Axis Length
	Shape Major Axis Length
	Shape Maximum 2D Diameter Row
	Shape Surface–Volume Ratio
	First-Order Minimum
	GLCM Correlation
	GLSZM Low-Gray-Level-Zone Emphasis
45	Shape Elongation
	Shape Least Axis Length
	Shape Major Axis Length
	Shape Maximum 2D Diameter Row
	Shape Surface–Volume Ratio
	First-Order Minimum
	GLCM Correlation
50	Shape Elongation
	Shape Least Axis Length
	Shape Major Axis Length
	Shape Maximum 2D Diameter Row
	Shape Surface–Volume Ratio
	First-Order Minimum
	GLCM Correlation
	GLRLM Run Entropy

## Data Availability

We confirm that the data supporting the findings of this article are available upon request from the corresponding author.

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
