# Peer review of "Radiomics in Dermatological Optical Coherence Tomography (OCT): Feature Repeatability, Reproducibility, and Integration into Diagnostic Models in a Prospective Study"

_cancers, 2025, doi:10.3390/cancers17050768_

Round 1
Reviewer 1 Report
Comments and Suggestions for Authors
The article is relevant and timely, addressing the application of radiomics in dermatological OCT. The study stands out by investigating the reproducibility and repeatability of radiomic features, a crucial aspect for the reliability of diagnostic models based on machine learning. However, the following points, if implemented, could contribute to better understanding and clarity of the proposed work:
Introduction
1) Expand the discussion on the impact of radiomics adoption in dermatology and the challenges of standardizing imaging acquisition protocols for better integration and acceptance as a support tool for specialists. Include more recent references (medical imaging and radiomics).
Methodology
2) Although validated by two physicians, manual segmentation may introduce bias. The use of semi-automatic methods could increase reproducibility. This limitation could be discussed.
3) The study's sample size (20 participants and 40 lesions) is relatively small. This limitation needs to be discussed more explicitly, reinforcing the necessity of external validation with larger datasets.
4) The 10-minute interval between scans was justified, but possible impacts of variables such as patient positioning or scanner pressure on image stability should be discussed.
5) Section 2.2 lacks details about the additional dataset used for classification. More information should be provided, and the sentence "Details in this..." should be replaced with a clearer explanation.
6) The dataset used is unbalanced. How was this issue addressed or mitigated in the classification task? This could be a critical factor for potential biases in analysis.
7) The CCC equation needs correction—both the equation and the explanation of its variables appear incorrect (verify the correct representation of the concordance correlation coefficient).
Results
8) Were the labels randomized in each epoch when the algorithm was executed? Wouldn’t this imply that, by the end of all epochs, the algorithm had seen all data, including learning from the test set? This could explain the high accuracy values and potential overtraining of the algorithm.
9) The authors could provide more details on how they prevented overfitting and overtraining of the algorithm.
Discussion
10) Strengthen the discussion on inherent differences between techniques (CT, MRI) and OCT when comparing the findings.
11) Regarding study limitations, the authors could further elaborate on the necessity of prospective validation in real clinical cohorts.
Author Response
Dear Reviewer,
We would like to take this opportunity to thank you for taking the time to review our manuscript. We found your comments and suggestions to be very insightful, and they have made a valuable contribution to strengthening our work. Please find our response to your comments, along with the changed sections in the manuscript. Furthermore, we have highlighted the revised text in yellow in the manuscript for your convenience.
Introduction
- Expand the discussion on the impact of radiomics adoption in dermatology and the challenges of standardizing imaging acquisition protocols for better integration and acceptance as a support tool for specialists. Include more recent references (medical imaging and radiomics).
Response: Thank you for your comment. We have updated the references to include more recent studies and have expanded the discussion on the challenges in the introduction. Specifically, we have added the following: [..In dermatology, for instance, the lack of standardized systems for storing and sharing dermatological images, unlike the Picture Archiving and Communication System (PACS) used in radiology, limits data interoperability[19] . Another major issue is feature stability, as some radiomics features are highly sensitive to slight variations in image acquisition, reconstruction parameters, or preprocessing steps …]. Line [ 83-88]
We have also included the following references:
Gust C, Schuh S, Welzel J, Daxenberger F, Hartmann D, French LE, Ruini C, Sattler EC. Line-field confocal optical coherence tomography increases the diagnostic accuracy and confidence for basal cell carcinoma in equivocal lesions: a prospective study. Cancers. 2022 Feb 21;14(4):1082.
Seoni S, Shahini A, Meiburger KM, Marzola F, Rotunno G, Acharya UR, Molinari F, Salvi M. All you need is data preparation: A systematic review of image harmonization techniques in Multi-center/device studies for medical support systems. Computer Methods and Programs in Biomedicine. 2024 Apr 23:108200.
Korecka K, Kwiatkowska D, Mazur E, DaÅ„czak-Pazdrowska A, Reich A, Å»aba R, PolaÅ„ska A. An Update on Non-Invasive Skin Imaging Techniques in Actinic Keratosis—A Narrative Review. Medicina. 2024 Jun 26;60(7):1043.
Nardone V, Reginelli A, Rubini D, Gagliardi F, Del Tufo S, Belfiore MP, Boldrini L, Desideri I, Cappabianca S. Delta radiomics: an updated systematic review. La radiologia medica. 2024 Aug;129(8):1197-214.
Bestetti A, Zangheri B, Gabanelli SV, Parini V, Fornara C. Union is strength: the combination of radiomics features and 3D-deep learning in a sole model increases diagnostic accuracy in demented patients: a whole brain 18FDG PET-CT analysis. Nuclear Medicine Communications. 2024 Jul 1;45(7):642-9.
Widaatalla Y, Wolswijk T, Adan F, Hillen LM, Woodruff HC, Halilaj I, Ibrahim A, Lambin P, Mosterd K. The application of artificial intelligence in the detection of basal cell carcinoma: A systematic review. Journal of the European Academy of Dermatology and Venereology. 2023 Jun;37(6):1160-7.
Methodology
- Although validated by two physicians, manual segmentation may introduce bias. The use of semi-automatic methods could increase reproducibility. This limitation could be discussed.
Response: Thank you for your comment. We acknowledge this concern and have now explicitly stated in the limitation part of the Discussion [……. Fourth, while our manual segmentation was validated by two assessors, it remains operator-dependent and may introduce variability. Semi-automatic methods could improve reproducibility, however, validated tools for OCT are limited. Future studies should explore automated approaches to reduce inter-observer variability.]. Line [ 390-394]
- The study's sample size (20 participants and 40 lesions) is relatively small. This limitation needs to be discussed more explicitly, reinforcing the necessity of external validation with larger datasets.
Response: Thank you for your comment. We have added this to the Discussion: […As our study establishes a framework for assessing feature robustness, external validation with larger datasets is needed to confirm reproducibility across different imaging settings and patient populations.]. Line [ 381-283]
- The 10-minute interval between scans was justified, but possible impacts of variables such as patient positioning or scanner pressure on image stability should be discussed.
Response: Thank you for your comment. We have addressed this in the Discussion by acknowledging how both probe pressure and patient positioning may contribute to acquisition-related variability.[..This can be attributed to the sensitivity of shape features to the geometry of the segmented lesion, which may vary due to slight differences in patient positioning and probe pressure during consecutive OCT scans.]. Line [ 315-318]
- Section 2.2 lacks details about the additional dataset used for classification. More information should be provided, and the sentence "Details in this..." should be replaced with a clearer explanation.
Response: Thank you for your comment. We have added the following information about the dataset in the Methods section: [..This dataset was obtained from a prospective cohort study at the Dermatology outpatient clinic of MUMC+, including patients who underwent skin biopsy for suspected non-melanoma skin cancer or premalignancy. Further details can be found in the work of Sinx et al. [24]…]. Line [ 136-139]
- The dataset used is unbalanced. How was this issue addressed or mitigated in the classification task? This could be a critical factor for potential biases in analysis.
Response: Thank you for your comment. We have addressed this in the Methods section by describing how class imbalance was handled: [Given that our dataset was imbalanced, with fewer cases of Bowen’s disease compared to BCC and benign nevi, we performed stratified 5-fold cross-validation to maintain class distribution across folds. Additionally, we evaluated model performance using AUC and precision-recall metrics, which provide more informative assessments for imbalanced datasets compared to accuracy alone.]. Line [ 200-204]
- The CCC equation needs correction—both the equation and the explanation of its variables appear incorrect (verify the correct representation of the concordance correlation coefficient).
Response: Thank you for your comment. We have corrected the CCC equation and its explanation in the Methods section to ensure accuracy. The updated equation follows the standard CCC formulation, and the variable definitions have been clarified: [The μ, ρ, and σ represent the mean, correlation coefficient, and variance for the two datasets (test and re-test), respectively. CCC values range from -1 (inverse concordance) to 0 (no concordance) and up to 1 (perfect agreement).]. Line [ 186-188]
Results
- Were the labels randomized in each epoch when the algorithm was executed? Wouldn’t this imply that, by the end of all epochs, the algorithm had seen all data, including learning from the test set? This could explain the high accuracy values and potential overtraining of the algorithm.
Response: Thank you for your comment. We have clarified that label randomization was applied once before training and remained unchanged during iterations, ensuring that the test set was never used during training. This has been added to the Methods section: […randomizing labels once before training … The test set remained separate throughout, and randomized labels were not altered during iterations.] .Line [ 213-216]
- The authors could provide more details on how they prevented overfitting and overtraining of the algorithm.
Response: Thank you for your comment. We have already detailed the steps used to prevent overfitting in the Methods section, including splitting the dataset into 70% training and 30% test sets, applying stratified 5-fold cross-validation, using regularization techniques , and feature selection.
Discussion
- Strengthen the discussion on inherent differences between techniques (CT, MRI) and OCT when comparing the findings.
Response: Thank you for your comment. We have addressed this issue and added it to the Discussion: [OCT, in comparison to CT and MRI, presents unique challenges in radiomics analysis. While CT benefits from standardized HU and MRI is affected by scanner-specific intensity variations, OCT lacks an established intensity normalization standard. Moreover, whereas CT and MRI radiomics features are influenced by slice thickness and reconstruction settings, OCT features are more sensitive to probe pressure, patient motion, and signal attenuation. These differences highlight the need for further research to standardize OCT radiomics and improve feature reproducibility for clinical applications.]. Line [ 306-312]
- Regarding study limitations, the authors could further elaborate on the necessity of prospective validation in real clinical cohorts.
Response: Thank you for your comment. We have addressed this in the Discussion: […Furthermore, we believe that prospective validation in real clinical cohorts is essential to evaluating the practical impact of incorporating OCT radiomics into routine dermatological workflows.]. Line [ 383-386]
Reviewer 2 Report
Comments and Suggestions for Authors
This paper describes the application of Radiomics processing to OCT images of skin cancers and benign skin conditions. Some potentially important conclusions are drawn, including that the method has sufficient promise for further development of automated diagnostic systems; and a useful recommendation for 'bin width' of 25/255 (~10% of total intensity range).
However, I do have some reservations about the work. The biggest one is the decision to use benign nevi as comparators for assessing performance. Benign nevi are easy to diagnose clinically; surely it would have been much better to have used benign lesions clinically difficult to differentiate from BCC, such as Actinic Keratoses that are clinically difficult to differentiate from Basal Cell Carcinoma? The use of benign nevi means that the model will be 'fitted' to distinguishing BCCs vs nevi instead of BCC vs AK, and therefore the derived AUC values are of questionable significance.
I presume that the benign nevi were not biopsied to histologically validate diagnosis, please confirm.
For a paper submitted to 'Cancers', there is surprisingly little detail on the target cancers (BCC, Bowen's), nor the motivation for developing automated diagnostic techniques. It is touched on in the Introduction, but I think more detail should be given to support the significance of the work: BCC is the most common cancer globally, and represents a significant burden on healthcare services and especially on overburdened histology services; the standard of care is to take a biopsy, leaving a scar and requiring a delay for the diagnosis; automated accurate non-invasive methods with instant feedback would be a significant benefit to patients, clinicians and healthcare costs. Providing more detail on this, with references, would more strongly underpin the relevance and motivation of this work.
Regarding the conclusion about bin width: I note that OCT imaging suffers from the phenomenon of Laser Speckle, which produces a localised variation of intensity at the optical resolution scale on the whole image; I suspect that Laser Speckle therefore has an impact on the bin width?
I was not clear from the method description, how exactly the AUCs were calculated. An OCT scan of a lesion actually comprises a 'stack' of image frames, typically 120 frames per scan for the VivoSight OCT device. Was the expert doctor segmentation performed on each and every image frame for each scan? When the Radiomics processing was performed on test scans, I suppose that the output was an automatically segmented set of image frames, and the AUC was calculated by counting the pixels that either matched or did not match the expert doctor's segmentation? Or have I not understood? Please clarify.
Some points about the manuscript:
- Some of the figures (2,3,4) have labels and text that is too small to read
- I do not understand Table 2? What is it showing?
- First paragraph of the Discussion appears to be a left over instruction to the authors
- There is a left over redundant heading '6. Patents'
In summary, I think this is useful work, but the paper should be improved to be more readable, it is hard to follow and lacks important contextual detail and explanation of the method, and it would be considerably strengthened in my opinion by using benign lesions that are clinically difficult to distinguish from BCC instead of benign nevi as comparator lesions.
Round 2
Reviewer 1 Report
Comments and Suggestions for Authors
The authors have improved the revised paper. In my opinion, it is now suitable for publication.
Reviewer 2 Report
Comments and Suggestions for Authors
The authors have addressed my concerns.